# Detection of Anti-SARS-CoV-2-S1 RBD-Specific Antibodies Prior to and during the Pandemic in 2011–2021 and COVID-19 Observational Study in 2019–2021

**DOI:** 10.3390/vaccines10040581

**Published:** 2022-04-10

**Authors:** Nadezhda G. Gumanova, Alexander U. Gorshkov, Natalya L. Bogdanova, Andrew I. Korolev, Oxana M. Drapkina

**Affiliations:** National Medical Research Center for Preventive Medicine of the Ministry of Healthcare of Russian Federation, 10 Petroverigsky per., Building 3, 101990 Moscow, Russia; agorshkov@gnicpm.ru (A.U.G.); nlbogdanova@gnicpm.ru (N.L.B.); akorolev@gnicpm.ru (A.I.K.); odrapkina@gnicpm.ru (O.M.D.)

**Keywords:** COVID-19, Sputink V, vaccination, seroprevalence, collective immunity

## Abstract

Background: Longitudinal surveys to monitor the seroprevalence are required to support efforts for assessment of the levels of endemic stability in certain countries. We investigated seroprevalence of anti-SARS-CoV-2-S1 receptor-binding domain (RBD)-specific antibodies in the serum samples in 2011–2021, including a cohort study of 2019–2021, to evaluate the vaccination and anti-IgG-SARS-CoV-2–S1 RBD-positive statuses to assess the resistance and severity of COVID-19. Materials and Methods: Anti-SARS-CoV-2-S1 RBD-specific antibodies were assayed in the serum samples (N = 565) randomly selected from various cohorts previously recruited from 2011 to 2021 from the city of Moscow and Moscow Region. Among them there were the participants (N = 310) recruited in 2019–2021 with an endpoint of 30 October 2021 when these participants were interviewed over phone with relevant questionnaire. Results: Obtained data indicated a percentage of 3–6% of SARS-CoV-2-S1 RBD-specific antibodies detected in participants recruited in 2011–2019. The percentage of SARS-CoV-2-S1 RBD-specific antibodies was increased to 16.5% in 2020 and to 46% in 2021. The vaccination rate of 238 respondents of this cohort was 58% from August 2020 to October 2021. In total, 12% of respondents were hospitalized. The morbidity rate in the subgroup of anti-SARS-CoV-2-S1 RBD-positive respondents was 5.4-fold higher than that in the subgroup of vaccinated respondents. Conclusions: A small percentage of SARS-CoV-2-S1 RBD-specific antibodies detected in 2011–2019 indicated possible spreading of coronaviruses during the pre-pandemic period. Collective immunity in Moscow and the Moscow region was able to reach 69% from August 2020 to October 2021 if this rate is added to the rate of not vaccinated SARS-CoV-2-S1 RBD-positive subjects.

## 1. Introduction

Over the past 20 years, world-wide aggressive coronavirus infections have been noted at least twice. In 2002–2003, 26 countries were affected by the SARS-CoV (severe acute respiratory syndrome coronavirus) epidemic, with a main peak in February 2003 and a lethality rate of 10% [1]. In 2012–2013, 27 countries were affected by MERS-CoV (Middle East respiratory syndrome coronavirus), with a lethality rate of up to 35% [2,3,4]. However, the violent global pandemic invasion of the coronavirus disease in 2019 (COVID-19 caused by SARS-CoV-2) was the most devastating. On 31 December 2019, the Chinese Center for Disease Control reported severe cases of pneumonia of unknown etiology in Wuhan. Subsequent identification determined that the causative agent is a novel Beta coronavirus (SARS-CoV-2). The robust data on the onset of SARS-CoV-2 infection and spreading of the virus in the pre-pandemic period are unavailable [5].

The profound impact of the COVID-19 pandemic indicates the need for additional investigations into multiple aspects of coronaviruses. Longitudinal surveys to continually monitor the seroprevalence around the globe are required to support the prevention and control efforts for assessment of the levels of endemic stability in certain countries and regions.

We investigated seroprevalence of anti-SARS-CoV-2-S1 receptor-binding domain (RBD)-specific antibodies in the samples of participants recruited for various cardiovascular studies in 2011–2018. A microcirculation cohort study recruited in 2019–2021 was used to evaluate the vaccination and anti-IgG-SARS-CoV-2–S1 RBD-positive statuses to assess the resistance and severity of COVID-19. The study on microcirculation was planned before the pandemic. Thus, exposure to the virus was apparently random and unrelated to the recruitment strategy.

## 2. Materials and Method

### 2.1. Participants

The present observational study included subjects 18 years of age and older randomly selected from various cohorts previously recruited from 2011 to 2021 from the city of Moscow and Moscow Region at the National Research Center for Preventive Medicine, Ministry of Health Care of Russian Federation, Moscow, Russia. All subjects were involved in various cardiovascular studies, and their recruitment was unrelated to occurrence of viral infection. The groups were selected from the corresponding cohorts (Figure 1) for a total of 565 serum samples. Recent participants (N = 310; recruited in 2019, N = 80; 2020, N = 132; and 2021, N = 98) were a part of a cohort for investigation of microcirculation (from August 2019 to August 2021).

The phone survey of the participants of this cohort was conducted in October 2021. The primary protocol for the 2019–2021 cohort was not aimed at investigating COVID-19, indicating that the participants were exposed to COVID-19 randomly. The groups recruited from 2011 to 2018 (N = 255) included 2011 (N = 30), 2012 (N = 31), 2013 (N = 22), 2014 (N = 45), 2015 (N = 45), 2016 (N = 20), 2017 (N = 27), and 2018 (N = 35). At the time of the recruitment visit, evaluation of the participants of the 2019–2021 cohort included assessment of baseline cardiovascular characteristics.

Anti-SARS-CoV-2-S1 RBD-specific antibodies were assayed in the serum samples (N = 565). At the endpoint (30 October 2021), participants of the 2019–2021 cohort (N = 310) were interviewed over phone using a questionnaire that included five questions as follows: (1) did the participant experience a respiratory disorder at any time before or after the enrolment in the study; (2) the date of the disorder; (3) severity of the disorder, including asymptomatic, mild, medium (fever and cough), or severe (hospitalization); (4) was the participant vaccinated and the date of vaccination; and (5) brand of a vaccine. 

The following exclusion criteria were used: any acute inflammation, including oral or dental; hematological diseases; left ventricular ejection fraction below 40%; diabetes mellitus; chronic kidney or liver failure; oncological diseases; mental illness; autoimmune diseases; any type of blood sugar-lowering therapy; any type of cholesterol-lowering therapy; pregnancy; and lactation. Patients with ischemic heart disease with an acute cardiovascular event less than 6 months before the enrollment were also excluded. The protocols of the study were approved by the local ethics committee and were compliant with the guidelines of the Helsinki Declaration and WHO. All participants gave their written informed consent for participation in the study with access to their personal data.

### 2.2. Antibody Assays

Blood was sampled from the ulnar vein after 12–14 h of fasting. The serum was aliquoted and stored at −76 °C.

Enzyme immunoassay for qualitative detection of IgG antibodies against SARS-CoV-2-S1 RBD was performed using an anti-SARS-CoV-2 ELISA E111-IVD kit (Mediagnost, Germany) by two-step enzyme-linked immunosorbent assay with recombinant SARS-CoV-2-S1 receptor binding domain (RBD) according to the manufacturer’s instructions. The samples were considered antibody-positive at >5-fold cut-off value or if the signal was higher than the optical density of 0.830. The samples with <3-fold cut-off value or with the signals lower than the optical density of 0.498 were considered antibody-negative. Samples from 3-fold to 5-fold cut-off or with the intermediate optical density values were considered borderline positive. The cut-off values were selected to ensure the exclusion of false positive results with a high probability. The assay was characterized by 95.55% sensitivity, 98.36% specificity, and 10.6% inter-assay variance.

The use of S1-RBD protein as an antigen in the assay enhances analytical specificity because the S1-RBD proteins of other human pathogenic coronaviruses share only 62–74% identity with SARS-CoV-2 [6]. However, cross-reactivity cannot be completely excluded due to relatively high similarity of the S1-RBD proteins of SARS-CoV-2 and SARS-CoV. Cross-reactivity of antibodies not specific to SARS-CoV-2 against SARS-CoV-2-S1 RBD protein in anti-SARS-CoV-2 ELISA E111-IVD was examined using the sera containing known antibodies against previously confirmed infections with other viruses. According to the data provided by the manufacturer, the ELISA kits did not cross-react with antibody-positive sera against Beta Corona HKU1, Beta Corona OC43, Alpha Corona 229E, and Alpha Corona NL63 confirmed by polymerase chain reaction tests and against SARS-CoV, varicella-zoster virus, hepatitis A, B, and C viruses, Epstein-Barr virus, cytomegalovirus, and herpes simplex virus.

Comparison with alternative commercially available assays indicates that the Mediagnost E111-IVD kits are well suited for the detection of anti-SARS-CoV-2 IgG antibodies. The S1 RBD viral antigen used as the binding target for antibodies is apparently superior to complete spike S1 protein or nucleocapsid protein, which both generate higher rates of false-positive results according to the data of the competitor assays [7].

### 2.3. Statistical Analysis

Statistical analysis was done using SPSS IBM version 23 software. OR with 95% CI data were used to compare the percentages of the participants with anti-SARS-CoV-2-S1 RBD-specific antibodies. Significance of Kaplan–Meier analysis was estimated with Log rank Mantel–Cox test. P values below 0.05 were considered significant.

## 3. Results

We investigated the presence of anti-SARS-CoV-2-S1 RBD-specific antibodies in the serum samples of 565 individuals enrolled in various cardiovascular studies from 2011 to 2021. All participants from 24 to 86 years of age (mean ± SD: 54 ± 9.2 years) were randomly selected from the three cohorts from 2011 to 2018 enrolled for various cardiovascular studies. A single cohort was recruited in 2019–2021 for a microcirculation study, which was not aimed to investigate viral infections. Thus, any contact with the viruses was apparently random for all participants. Moreover, exclusion criteria included acute systemic inflammation and oral or dental inflammation.

IgG antibodies against SARS-CoV-2-S1 RBD protein in the serum were assayed in 565 participants of all groups recruited from 2011 to 2021 (Table 1, Figure 2). 

The signals corresponding to IgG antibodies against SARS-CoV-2-S1 RBD were detected in 1 out of 30 participants in 2011, 2 out of 31 participants in 2012, 1 out of 22 participants in 2013, 3 out of 45 participants in 2014, 2 out of 45 participants in 2015, 1 out of 20 participants in 2016, 0 out of 27 participants in 2017, 1 out of 35 participants in 2018, 4 out of 80 participants in August–December 2019, 22 out of 132 participants in 2020, and 45 out of 98 participants enrolled from December 2020 to August 2021 (Table 1, Figure 2). The data indicated that IgG-positive samples with anti-SARS-CoV-2-S1 RBD antibodies were present in 3–6% of participants enrolled from 2011 to 2019. The fraction of IgG-positive participants was increased to 16.5% in 2020 and to 46% in 2021. The cohort enrolled in 2019–2021 was analyzed based on months. The number of participants and percentage of IgG-positive samples in the groups are presented in Table 2. The distribution of IgG-positive samples in 2019–2021 demonstrated that the fraction of anti-SARS-CoV-2-S1 RBD-positive samples was increased from April 2021 to August 2021 compared to that in January–March 2021. The fraction of SARS-CoV-2-S1 RBD-positive samples was increased up to 75% in August 2021, when enrolment in this cohort has ended (Table 2, Figure 3). 

Anti-SARS-CoV-2-S1 RBD-specific antibodies were detected in 70 (22.5%) participants of the cohort enrolled in 2019–2021 (N = 310) (Table 3). Subgroups of participants with anti-SARS-CoV-2-S1 RBD-positive and -negative statuses had similar clinical, demographic, and biochemical parameters, including sex, age, systolic blood pressure, diastolic blood pressure, body mass index, waist circumference, lipid profile, fibrinogen, blood glucose, and C-reactive protein levels (data not shown). The participants of the 2019–2021 cohort were subsequently followed up by phone survey starting from 30 October 2021. Overall, 251 out of the enrolled 310 (81%) participants were possible to follow up by phone survey, with 13 participants (4.2%) refusing to answer. Those participants who were able to answer the questions of the survey were designated respondents, including 93 out of 238 (39%) respondents who reported any symptoms of respiratory disorders. These symptoms were classified into mild (47%), medium (cough, cold, and fever) (41%), and sever (hospitalization) (12%) cases. COVID-19 was verified by PCR-test in 91% of 93 participants who reported any symptoms (Table 3). 

According to the data of the follow up survey, 137 (58%) out of 238 respondents were vaccinated from September 2020 to October 2021, including 87% of respondents who were able to recall the information, were vaccinated with Gam-COVID-VAC (Sputnik V) (N = 119, 87%) (Table 3). A total of 14 respondents of the 2019–2021 cohort were vaccinated before the enrolment, and anti-SARS-CoV-2-S1 RBD–specific antibodies were detected in 64% of these 14 individuals. Thus, 9 out of 70 IgG-positive respondents of the 2019–2021 cohort could have developed specific IgG antibodies due to vaccination, and the remaining 61 out of 310 (20%) of IgG-positive respondents could have developed IgG antibodies due to COVID-19. Notably, time intervals between vaccination and recruitment visit in these two subgroups were significantly different; the corresponding values were 9–151 days for anti-IgG-SARS-CoV-2-S1 RBD–positive respondents and 117–169 days for anti-IgG-SARS-CoV-2-S1 RBD-negative respondents (*p* < 0.05) (Table 3). The differences in seropositivity of these subgroups can be due to the duration of immunity, which is estimated to be 3–4 months, or due to efficacy of the vaccine. 

All respondents (N = 238) were divided into four partly overlapping subgroups, including vaccinated (subgroup A, N = 137), anti-SARS-CoV-2-S1 RBD-positive (subgroup B, N = 59), not vaccinated (subgroup C, N = 97), and anti-SARS-CoV-2-S1 RBD-negative (subgroup D, N = 179). The Kaplan–Meier analysis was performed after a maximum of 2.5-year follow-up period to determine the differences in the relapse of COVID-19 between these subgroups (Table 4, Figure 4). 

The Kaplan–Meier curves based on the relapse outcome are shown in Figure 4, indicating that the vaccinated subgroup (87% Sputnik) was less likely to get sick with COVID-19 compared to the subgroups B, C, and D according to the Log rank test (*p* = 0.001; Table 4). Unexpectedly, the subgroup B of anti-SARS-CoV-2-S1 RBD-positive respondents had no COVID-19 resistance compared with that of the anti-SARS-CoV-2-S1 RBD-negative (D) and not vaccinated (C) subgroups (*p* > 0.05; Log rank test). The data indicated that 20% of anti-SARS-CoV-2-S1 RBD-positive and 4% of vaccinated respondents had a relapse of COVID-19 within 10–629 days after the recruitment visit date and 92–183 days after vaccination, respectively. The morbidity rate in the subgroup of anti-SARS-CoV-2-S1 RBD-positive respondents was 5.4-fold higher than that in the subgroup of vaccinated respondents (OR = 5.4; CI: 1.4–19.8; *p* = 0.01; Table 3). Notably, the subgroup of anti-SARS-CoV-2-S1 RBD-negative respondents was less likely to get sick with COVID-19 compared to the subgroup of not vaccinated (*p* = 0.001; Log rank test). Apparently, the respondents who were not sick with COVID-19 had stronger overall immunity. However, IgG-positive and -negative participants of the 2019–2021 cohort had similar levels of various biochemical parameters, including C-reactive protein, and anthropometric and clinical parameters (data not shown).

Thus, the data of the present study indicated that a small percentage (3–6%) of SARS-CoV-2-S1 RBD-specific antibodies was detected in participants recruited in 2011–2019. The percentage of SARS-CoV-2-S1 RBD-specific antibodies was increased to 16.5% in 2020 and to 46% in 2021. The vaccination rate of 238 respondents of this cohort was 58% from August 2020 to October 2021. Combining the number of vaccinated (58%) and unvaccinated anti-SARS-CoV-2-S1 RBD-positive subjects (11%) yielded a possible estimate for the collective immunity rate (69%). 

The data indicated that 12% of respondents who reported any symptoms of respiratory disorders (N = 93) were hospitalized. No cases of hospitalization were reported in vaccinated respondents. No deaths were reported in respondents who participated in the follow up phone survey.

## 4. Discussion

Questions related to the origin of coronavirus remain on scientific agenda. Hence, the actual time frame of the onset of severe acute respiratory syndrome due to SARS-CoV-2 infection and possible spreading of coronaviruses during the pre-pandemic period are of great interest. SARS-CoV-2 seroprevalence had considerable geographic variability, as expected in the case of a pandemic [8]. In addition, anti-SARS-CoV-2 RBD-specific antibodies were detected in Italy several months before the first COVID-19 patient was identified. These findings have a potential to redefine the timeframe for the onset of COVID-19 pandemic with detection anti-SARS-CoV-2 RBD-specific antibodies in 11.6% of participants recruited from September 2019 to August 2021 [9]. 

The results of an internet search carried out in Italy using Google Trends (Google Inc., Mountain View, CA, USA) with the most commonly reported COVID-19 symptoms in Italy indirectly supports the conclusions of Apolone et al. [9] that SARS-CoV-2 may have started to circulate in Italy in September 2019, specifically in Lombardy, before the declared pandemic period [10]. 

The results from multiple independent assays support these observations, demonstrating the presence of preexisting antibodies, which were able to recognize SARS-CoV-2, in uninfected individuals [11,12,13]. These findings corresponded with the results of the present study on the presence of anti-SARS-CoV-2-S1 RBD-specific antibodies in 5% of participants recruited in August–December 2019. 

Cross-reactivity between SARS-CoV-2 and other circulating coronaviruses can explain our observations. Almost 90% of the human population is seropositive for at least three known persistently circulating coronaviruses, such as HCoV-OC43 (human coronavirus OC43), HCoV-HKU1, HCoV-NL63, and HCoV-229E, which cause certain respiratory disorders [12]. A study demonstrated ELISA cross-reactivity of SARS-related coronaviruses, MERS, and SARS-CoV sera only with the full SARS-CoV-2 S protein, but not with the S1 antigen, which is in agreement with several other studies that failed to detect cross-reactivity of anti-SARS-related coronavirus antibodies with SARS-CoV-2 RBD. Cross-neutralization activity between convalescent sera from SARS and COVID-19 patients was shown to be limited [14]. Moreover, the lack of cross-reactivity between the S1 subdomain of SARS-CoV-2 and other SARS-associated coronaviruses was used to develop a target antigen for highly specific serological assays [4,5].

Serologic surveys are valuable tools for the investigation of public health; however, these surveys are able to assess the presence of an infection only indirectly and are less reliable than standard diagnostic methods. ELISAs specific for the detection of anti-SARS-CoV-2 antibodies for RBD, S1, and N proteins have sensitivities of 88.4%, 89.3%, and 72.9%, respectively, and specificity of these ELISAs is over 94% [15]. According to manufacturer characteristics, ELISA kits used in the present study provide diminished cross-reactivity with other coronaviruses (Materials and Methods).

The presence of anti-SARS-CoV-2-S1 RBD-specific antibodies in the samples in 2011–2018 can be due to long-term humoral immunity induced by previous infections of coronaviruses, including SARS-CoV or MERS-CoV. In general, IgG antibodies appear at a later stage of immune response because IgGs undergo affinity maturation through somatic mutations, resulting in high affinity against a target antigen and enhanced capacity to neutralize the pathogens [4,5]. A long-lasting immunity (up to 6 years) against these viruses has been demonstrated previously [4,5].

Obtained results corresponded with previous observations by León et al., demonstrating that vaccination protects against COVID-19 and related hospitalization [16]. 

## 5. Conclusions

The present study detected anti-SARS-CoV-2-S1 RBD-specific antibodies (3–6%) in the serum of participants recruited in 2011–2019, and these findings can contribute to our understanding of the origins of SARS-CoV-2. The levels of anti-SARS-CoV-2-S1 RBD-specific antibodies were increased to 16.5% in 2020 and to 46% in 2021. From August 2020 to October 2021, 58% of the respondents have been vaccinated (87% Sputnik). Apparently random encounters of respondents the 2019–2020 cohort with COVID-19 can represent overall vaccination levels in general population in Moscow, Russia. Combining the number of vaccinated (58%) and unvaccinated anti-SARS-CoV-2-S1 RBD-positive subjects (11%) yielded a possible estimate for the collective immunity rate (69%). Vaccinated respondents had enhanced Covi-19 resistance, whereas anti-SARS-CoV-2-S1 RBD-positive respondents did not demonstrate any advantageous COVID-19 resistance. Disease resistance in vaccinated respondents was five-fold higher than that in the group of not vaccinated respondents, and no COVID-19-related hospitalizations were reported in vaccinated respondents. In the case of not vaccinated respondents, hospitalizations were reported in 12% of the cases associated with any symptoms of respiratory disorders. No deaths were reported in respondents. Thus, specific vaccination against SARS-CoV, but not immunity acquired via infection, provides significant protection against subsequent COVID-19.

## 6. Limitations

The main limitation of the present study is the survey-based follow up in the 2019–2021 cohort. Notably, the follow up response rate in the present study was relatively high at 81%. Thus, any potential variability can be predominantly due to recall bias of the respondents, which has been reported as substantial for unrepresentative surveys [17]. However, the phone survey used as follow up in the present study was performed by qualified study personnel, and the participants were highly likely to answer the follow up questions accurately due to potential relevance of these questions to the parental microcirculation study. Another limitation is that the dynamics of morbidity in the 2019–2021 cohort illustrated in Figure 4 may suggest insufficient follow-up period and number of respondents, requiring additional confirmatory investigations in the future.

## Figures and Tables

**Figure 1 vaccines-10-00581-f001:**
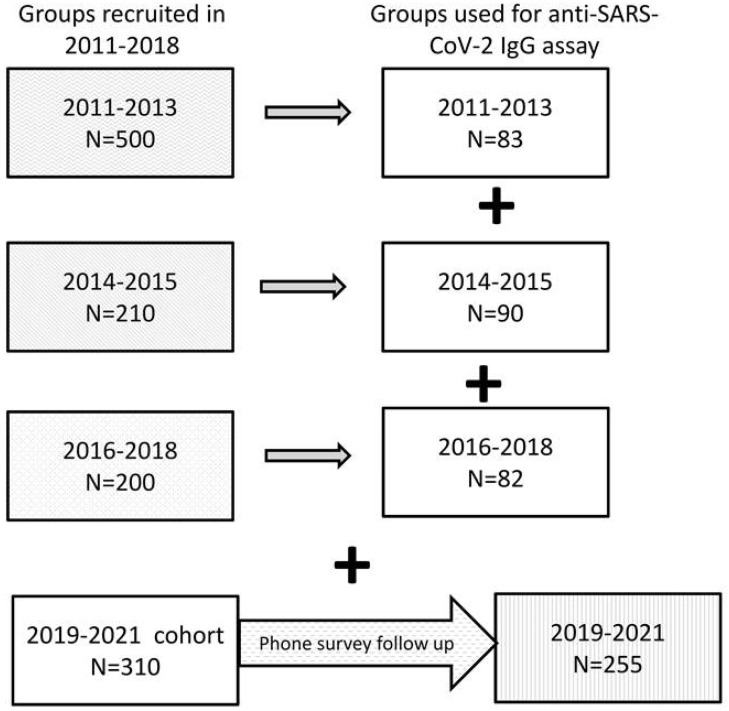
Protocol of the study. The present study was based on various cohorts previously recruited from 2011 to 2021 in the city of Moscow and Moscow Region. Subjects involved in various cardiovascular studies of the groups 2011–2013 and 2016–2018 represented patients with coronary artery disease, and apparently healthy individuals were included in the cohort study of microcirculation (groups 2014–2015 and 2019–2021). The groups were selected from the corresponding cohorts. The phone survey of the participants of cohort 2019–2021 was conducted in October 2021. The primary protocol for the 2019–2021 cohort was not aimed at investigating COVID-19. All participants were exposed to COVID-19 randomly.

**Figure 2 vaccines-10-00581-f002:**
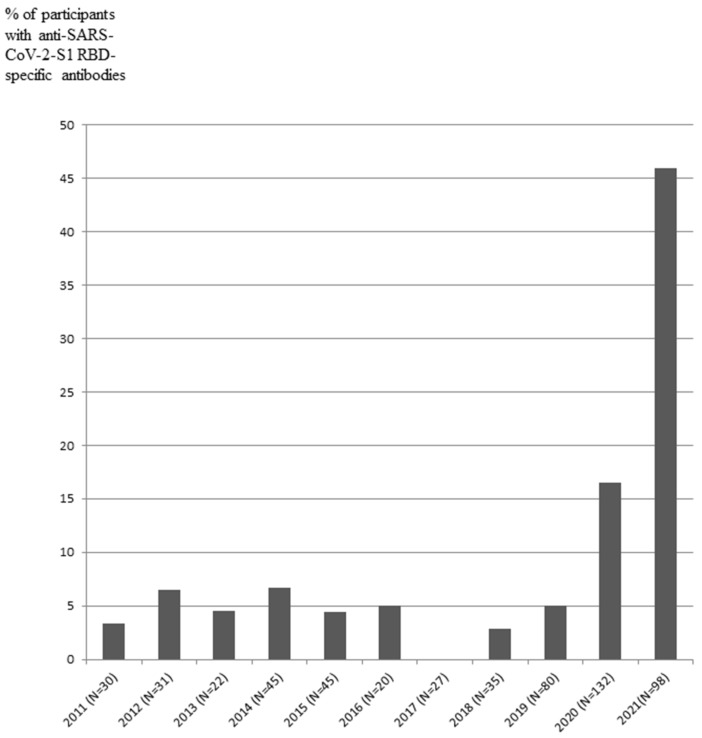
Detection of anti-SARS-CoV-2-S1 RBD-specific antibodies (% of participants) in the period before the pandemic 2011–2019 and during the pandemic from 2020 to 2021. No differences in % 2011–2019, *p* > 0.05; No differences in % 2020 vs. 2011–2019, *p* = 0.01; Differences in % 2021 vs. 2020, R = 2.7, % CI:1.5 to 4.9, *p* = 0.0005.

**Figure 3 vaccines-10-00581-f003:**
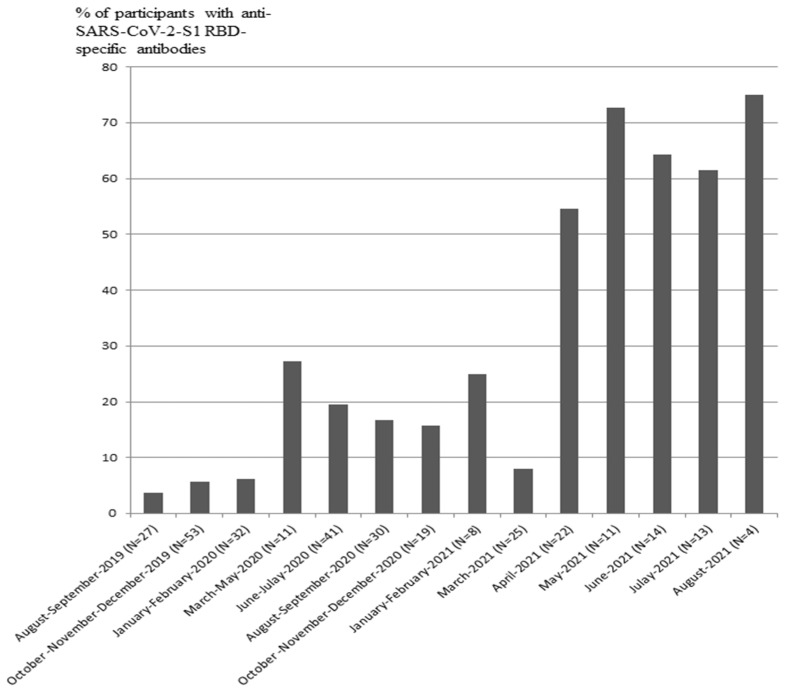
Detection of anti-SARS-CoV-2-S1 RBD-specific antibodies (% of participants) in the pandemic period from 2019 to 2021. OR (April 2021–August 2021 vs. August 2019–March 2021) = 4.1; 95% CI 1.05–16.4; *p* = 0.04).

**Figure 4 vaccines-10-00581-f004:**
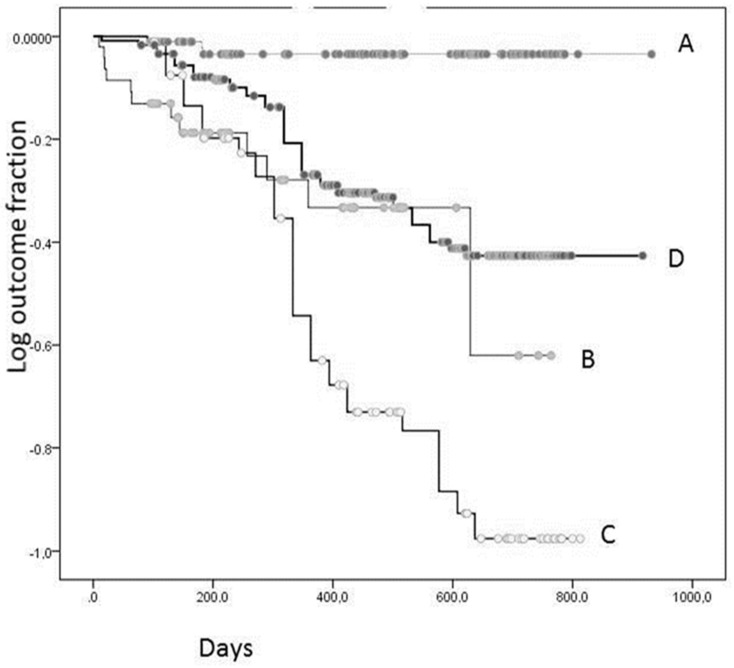
Kaplan-Meier analysis for COVID-19 relapse outcome in various groups of respondents: A, vaccinated; B, IgG-positive; C, not vaccinated; and D, IgG-negative.

**Table 1 vaccines-10-00581-t001:** Participants analyzed based on years from 2011 to 2021 and serum anti-SARS-CoV-2 IgG antibody status.

Year-Based Groups	Total N and Age (Mean ± SD)	Year	N for Indicated Years	Serum Anti-SARS-CoV-2 (S1-RBD) IgG Antibody-Positive (N; %)
2011–2013	83; 61.2 ± 9.4	2011	30	1 (3.3)
		2012	31	2 (6.5)
		2013	22	1 (4.5)
2014–2015	90; 53.6 ± 7.4	2014	45	3 (6.7)
		2015	45	2 (4.4)
2016–2018	82; 58.1 ± 11.1	2016	20	1 (5.0)
		2017	27	0 (0.0)
		2018	35	1 (2.9)
2019–2021	310; 42 ± 8.7	2019 (from August to December)	80	4 (5.0)
		2020	132	22 (16.5)
		2021 (from January 2021 to August 2021)	98	45 (45.9)

**Table 2 vaccines-10-00581-t002:** Participants (N and %) with serum anti-SARS-CoV-2 (S1) IgG-positive status based on indicated months in the 2019–2021 group, OR (August 2019–March 2021 vs. April 2021–August 2021) = 4.1; 95% CI 1.05–16.4; *p* = 0.04).

Months	Participants (N)	Participants with Serum Anti-SARS-CoV-2-S1 IgG-Positive Status (%)
August–September 2019	27	3.7
October–December 2019	53	5.7
January–February 2020	32	6.3
March–May 2020	11	27.3
June–July 2020	41	19.5
August–September 2020	30	16.7
October–December 2020	19	15.8
January–February 2021	8	25.0
March 2021	25	8.0
April 2021	22	54.5
May 2021	11	72.7
June 2021	14	64.3
July 2021	13	61.5
August 2021	4	75

**Table 3 vaccines-10-00581-t003:** COVID-19 infections in respondents enrolled in 2019–2021.

Description of Participants/Respondents	Respondents/Total Participants or Respondents (N/N; %)	IgG-Positive Respondents (N/N; %)	IgG-Negative Respondents (N/N; %)
Participants			
All participants, 2019–2021	310 (100)	70/310 (22.5)	240/310 (77.5)
Available for phone survey	251/310 (81)		
Agreed to answer (respondents)	238/310 (77)	59/310 (19)	179/310 (58)
Refused to answer	13/310 (4)		
Unavailable for phone survey	59/310 (19)		
Respondents			
Symptoms and severity of respiratory disorders	238 (100)		
Any symptoms reported	93/238 (39)	30/59 (37)	63/179 (35)
No symptoms reported	145/238 (61)	40/59 (63)	105/179 (65)
COVID-19 verified by PCR	85/93 (91)	30/93 (32)	55/93 (59)
Severity of respiratory disorders			
Mild	44/93 (47)	15/30 (50)	28/63 (44)
Medium (cough, cold, and fever)	38/93 (41)	12/30(40)	26/63 (41)
Severe (hospitalization)	11/93 (12)	3/30 (10)	9/63 (14)
Vaccination status	238 (100)	59 (100)	179 (100)
Vaccinated (from September 2020 to October 2021)	137/238 (58)	33/59 (56)	104/179 (58)
Not vaccinated	97/238 (40)	25/59 (42)	72/179 (40)
Cannot answer	4/238 (2)	1/59 (2)	3/179 (2)
Brand of vaccine in respondents who were able to answer	119 (100)		
Gam-COVID-Vac (Sputnik V)	103/119 (87)		
Other vaccines:	16/119 (13)		
CoviVac	7/119 (6)		
EpiVacCorona	5/119 (4)		
Sputnik Lite	4/119 (3)		
Vaccinated before enrolment, IgG-status (Gam-COVID-Vac, N = 13; CoviVac, N = 1)	14/238 (6)	9/14 (64)	5/14 (36)
**Days from vaccination to the enrollment visit**		**9–151 ***	**117–169 ***
COVID-19 disease status	238 (100)	59 (100)	179 (100)
COVID-19 verified with PCR-tests	94/238 (40)	27/59 (46)	66/179 (37)
Not verified	144/238 (60)	31/59 (54)	113/179 (63)
IgG-positive respondents who were able to recall the date of respiratory disorders		59 (100)	
Reported a relapse of respiratory disorders after the enrollment visit during 10–629 days of follow-up		**12/59 (20) ****	
Reported a respiratory disorder before the enrollment visit during 27–476 of days follow-up		21/59 (36)	
COVID-19 verified by PCR-tests		16/21(76)	
No respiratory disorders reported		26/59 (44)	
Asymptomatic		2/21 (10)	
Mild		10/21(48)	
Medium (cough, cold, and fever)		8/21 (38)	
Severe (hospitalization)		1/21 (5)	
Severity grade of respiratory disorders after the enrollment visit in IgG -positive respondents who were able to answer		12 (100)	
COVID-19 verified by PCR-tests		12/12 (100)	
Asymptomatic		1/12(8)	
Mild		6/12 (50)	
Medium (cough, cold, and fever)		4/12 (33)	
Severe (hospitalization)		1/12 (8)	
All vaccinated respondents	137 (100)		
Vaccinated respondents who were able to recall the date of respiratory disorders	118/137 (86)		
Respondents with reported respiratory disorders before vaccination	39/118 (33)		
Respondents with reported relapse of COVID-19 in vaccinated subjects verified by PCR-tests (from 92 to 183 days after the vaccination; mean 152 days)	**3/79 (4) ****	1/3 (33.3)	2/3 (66.7)
Severity grade of relapse respiratory disorders in vaccinated respondents			
Mild	2/79 (3)		
Medium (cough, cold, and fever)	1/79 (1)		
Severe (hospitalization)	-		

(Bold) Differences in days from vaccination to the enrollment visit * *t*-test, *p* < 0.05. A relapse of respiratory disorders after the enrollment visit during ** Odds ratio = 5.4 (CI: 1.4–19.8, *p* = 0.01) for IgG-positive respondents with reported relapse of respiratory disorders versus vaccinated respondents who reported relapse of COVID-19.

**Table 4 vaccines-10-00581-t004:** Pairwise comparisons of the groups of respondents with various vaccination and IgG statuses according to Kaplan–Meier analysis (Log rank Mantel–Cox test).

Groups	Vaccinated		IgG-Positive		Not Vaccinated		IgG-Negative	
	Chi Squared Test	*p*	Chi Squared Test	*p*	Chi Squared Test	*p*	Chi Squared Test	*p*
Vaccinated			20.75	0.001	51.20	0.001	20.93	0.001
IgG-positive	20.75	0.001			1.59	0.21	1.31	0.25
Not vaccinated	51.20	0.001	1.59	0.21			19.74	0.001
IgG-negative	20.93	0.001	1.30	0.25	19.73	0.001		

## Data Availability

Data available on request due to restrictions privacy or ethical.

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
