# Peer review of "Detection of Anti-SARS-CoV-2-S1 RBD-Specific Antibodies Prior to and during the Pandemic in 2011–2021 and COVID-19 Observational Study in 2019–2021"

_vaccines, 2022, doi:10.3390/vaccines10040581_

Round 1
Reviewer 1 Report
Gumanova et al. looked for the presence of antibodies against SARS-CoV-2-S1 receptor-binding domain (RBD)–specific in the serum samples from two cohorts of patients followed in the context of cardiovascular studies between 2011 and 2019 and between 2019 and 2021. They found that the presence of antibodies in a minority of few patients before 2020.
I have several questions:
- It is not clear to me what the authors mean by “subjects involved in various cardiovascular studies”, are they healthy individuals with preventive cardiovascular test or are they patients treated for cardiovascular diseases ?
- The ELISA test is interested in the antibodies agains the S protein of the virus, patients positive for the anti-S, were they tested too for the anti nucleoprotein of Sars-Cov2 ?
- ELISA test is said to be positive or negative. For the first cohort, what are the titers for patients with positive tests? Are they lower or similar to those of vaccinated or infected patients ?
- If I read correctly, individuals with positive ELISA prior to pandemic and nor vaccinated were not protected against Covid19. Is it correct ?
Reviewer 2 Report
The manuscript titled “Detection of anti-SARS-CoV-2 S1 RBD-specific antibodies prior to and during the pandemic in 2011-2021 and Covid-19 observational study in 2019-2021” by Gumanova et al. examined seroprevalence of anti-SARS-CoV-2-S1 receptor-binding domain (RBD)–specific antibodies in the serum samples including 310 cohort samples to evaluate the vaccination and anti-IgG-SARS-CoV-2–S1 RBD-positive statuses and assessed the resistance and severity of Covid-19 in a regional population from the city of Moscow area. The questions examined are very interesting: anti-SARS-CoV-2 S1 RBD-specific antibodies are detectable in this regional population before the COVID-19 pandemic and the collective immunity significantly increased during COVID-19 pandemic despite the vaccination. The data collection and data presentations are well organized . There are only a few minor suggestions listed below if it’s helpful to improve the manuscript:
- The statistical analysis section has insufficient details and the differences between percentages can be statistically tested to avid the percentage difference arisen by chance (Fig2 and 3).
- “Collective immunity in Moscow and Moscow region was able to reach 69% from August, 2020 to October, 2021 if this rate is added to the rate of not vaccinated SARS-CoV-2-S1 RBD-positive subjects”. Consider to rewrite this sentence to make the conclusion more clear., e.g., look at data by excluding those vaccinated subjects?
- Line 18-19: ‘including participants (N = 310); recruited in 2019-2021 and endpoint October 30, 2021, participants of the 2019-2021 cohort (N = 310) were interviewed over phone with relevant questionnaire”. This sentence is easily confusing with 2 “(N=310)”, suggest to re-organize it.
- It would be better to add y axis titles to Figure 2 and 3 and increase their resolutions.
- Suggest to make the column 3 in the table 2 similar to the column 5 in the table 1, something like (N, %). Check number in the last row of table 2, which is likely error.
- Table 1, last row in column 3: time is overlapping, suggested to make it clear.
- Table 3 with 2 ** Odds ratio tests but there are only parameters for 1 test in the table annotation for “IgG-positive respondents with reported relapse of respiratory disorders versus vaccinated respondents who reported relapse of Covid-19”.
- Suggest increase the resolution of Fig4 and make y text label with the same digits and x text label with integer for days and add x axis title.
- Suggest to make Figure legends more clear, especial for Figure 1.
- Line 183: extra space; Line 321: missing space.
- Reference format needs to be checked.
